# Insight into genetic predisposition to chronic lymphocytic leukemia from integrative epigenomics

Helen E. Speedy[1,10], Renée Beekman [2,3,10], Vicente Chapaprieta [4], Giulia Orlando[1], Philip J. Law [1], David Martín-García[2,3], Jesús Gutiérrez-Abril[5], Daniel Catovsky[1], Sílvia Beà [2,3], Guillem Clot [2,3], Montserrat Puiggròs[6], David Torrents[6,7], Xose S. Puente [3,5], James M. Allan[8], Carlos López-Otín [3,5], Elias Campo[2,3,4,9], Richard S. Houlston [1,11] & José I. Martín-Subero[2,3,4,7,11]

Genome-wide association studies have provided evidence for inherited genetic predisposition to chronic lymphocytic leukemia (CLL). To gain insight into the mechanisms underlying CLL risk we analyze chromatin accessibility, active regulatory elements marked by H3K27ac, and DNA methylation at 42 risk loci in up to 486 primary CLLs. We identify that risk loci are significantly enriched for active chromatin in CLL with evidence of being CLL-specific or differentially regulated in normal B-cell development. We then use in situ promoter capture Hi-C, in conjunction with gene expression data to reveal likely target genes of the risk loci. Candidate target genes are enriched for pathways related to B-cell development such as MYC and BCL2 signalling. At 14 loci the analysis highlights 63 variants as the probable functional basis of CLL risk. By integrating genetic and epigenetic information our analysis reveals novel insights into the relationship between inherited predisposition and the regulatory chromatin landscape of CLL.

[1] Division of Genetics and Epidemiology, Institute of Cancer Research, London SW7 3RP, UK. [2] Institut d'Investigacions Biomèdiques August Pi i Sunyer (IDIBAPS), 08036 Barcelona, Spain. [3] Centro de Investigación Biomédica en Red de Cáncer (CIBERONC), 28029 Madrid, Spain. [4] Departament de Fonaments Clinics, Facultat de Medicina, Universitat de Barcelona, 08036 Barcelona, Spain. [5] Departamento de Bioquímica y Biología Molecular, Instituto Universitario de Oncología (IUOPA), Universidad de Oviedo, 33006 Oviedo, Spain. [6] Programa Conjunto de Biología Computacional, Barcelona Supercomputing Center (BSC), Institut de Recerca Biomèdica (IRB), Spanish National Bioinformatics Institute, Universitat de Barcelona, Barcelona, Spain. [7] Institució Catalana de Recerca i Estudis Avançats (ICREA), 08010 Barcelona, Spain. [8] Northern Institute for Cancer Research, Newcastle University, Newcastle upon Tyne NE2 4HH, UK. [9] Hematopathology Section, Hospital Clinic of Barcelona, 08036 Barcelona, Spain. [10] These authors contributed equally: Helen E. Speedy, Renée Beekman. [11] These authors jointly supervised this work: Richard S. Houlston, José I. Martín-Subero. Correspondence and requests for materials should be addressed to R.S.H. (email: richard.houlston@icr.ac.uk) or to J.I.M.-S. (email: imartins@clinic.cat)

Chronic lymphocytic leukemia (CLL) is an indolent B-cell malignancy that has a strong genetic component, as evidenced by the eight-fold increased risk in relatives of CLL patients[1]. Our understanding of CLL predisposition has been transformed by genome-wide association studies (GWAS), which have identified alleles at 43 loci influencing risk[2–9]. Elucidating the function of these risk loci is an important step toward the development of testable hypotheses regarding the biological processes involved in CLL predisposition and pathogenesis. Most GWAS signals, however, map to the non-coding genome and elucidating the mechanisms through which these non-coding variants exert their effect has proven challenging[10]. Integration of chromatin immunoprecipitation–sequencing (ChIP-seq), chromatin accessibility, DNA methylation, and gene expression profiles with GWAS data has revealed novel biological insights for other diseases[11]. Thus far, understanding the molecular basis of CLL susceptibility loci has been limited because genome-wide information on the CLL regulome has not been available for a large CLL series.

In this study, we sought to address this deficiency by analyzing a large, genetically and epigenetically well-characterized CLL series, together with epigenome data of the normal B-cell lineage[12–14]. First, by characterizing the epigenomes of CLL and different stages of B-cell differentiation we gained insight into the developmental basis of CLL risk. Second, we explored the CLL risk loci using histone modification profiles, chromatin accessibility assessed by assay for transposase-accessible chromatin using sequencing (ATAC-seq), DNA methylation, and gene expression data in conjunction with DNA genotypes in the same CLL cases. Furthermore, we used three-dimensional chromatin data to explore physical interactions between the risk loci and their candidate target genes. These analyses allowed for a detailed interpretation of GWAS signals for CLL and to infer the molecular mechanisms through which loci operate.

## Results

**The active chromatin landscape at CLL risk loci.** The majority of GWAS risk loci map to non-coding regions of the genome and influence gene regulation[10]. Hence, to gain insight into the biological basis underlying genetic CLL predisposition, we evaluated profiles of three histone marks related to active regulatory elements at each locus in CLL cells (determined by H3K27ac, H3K4me3, and H3K4me1 ChIP-seq marks). We first considered each of the 42 non-HLA risk loci solely on the basis of the strongest single-nucleotide polymorphism (SNP) association with CLL in our recent meta-analysis[2] (Supplementary Data 1). One third (14/42) of the risk SNPs localize to either active promoters or enhancers in CLL cells (Fig. 1a). Moreover, on the basis of linkage disequilibrium (LD; $r^2 \geq 0.2$) with the sentinel SNP, 93% (39/42) of the risk loci featured SNPs mapping to an active regulatory element (Fig. 1a and Supplementary Data 2; $P = 1.3 \times 10^{-8}$, Fisher's exact test). Specifically, on average 81% (34/42) of the LD-defined CLL risk loci showed enrichment of active promoters and/or enhancers per CLL sample (Fig. 1b, Supplementary Data 3). This enrichment of CLL-related active regulatory elements at CLL risk loci contrasts with the significantly lower enrichment observed at both colorectal (CRC)[15] and breast cancer (BC)[16] GWAS loci (Fig. 1b, Supplementary Data 3). Chromatin state enrichment in CLL essentially defines five groups of risk loci characterized by: (i) active regulatory elements only ($n = 15$ loci), (ii) active regulatory elements and transcribed regions ($n = 10$ loci), (iii) weak regulatory elements only ($n = 10$ loci), (iv) weak regulatory elements and transcribed regions ($n = 3$ loci), and (v) poised and inactive regions ($n = 4$ loci) (Fig. 1a).

Since the epigenome of B cells varies according to their maturation state[12,17], we examined H3K27ac profiles at the risk loci in CLL and a range of normal B-cell subpopulations (Fig. 1c, Supplementary Data 4). H3K27ac signals at 10 risk loci (represented by 30 H3K27ac peaks) were significantly increased in CLL cells as compared to normal B-cell subpopulations, a significant higher number than expected ($P = 3.5 \times 10^{-3}$, Fisher's exact test, 11 peaks based on 1000 permutations). These included five regions of de novo activation previously documented in CLL[12] (Supplementary Data 5). Overall, 93% of the LD-defined risk loci (39/42, represented by 282 H3K27ac peaks) displayed differential H3K27ac profiles across CLL and the different normal B-cell subpopulations, also significantly more than that expected ($P = 3.0 \times 10^{-3}$, Fisher's exact test, 239 peaks based on 1000 permutations). Collectively, our data show that CLL risk loci are enriched for CLL-related regulatory elements being CLL specific or having differential regulation across CLL and B-cell development, as exemplified by the loci marked by the SNPs rs11637565 and rs35923643 (Fig. 1d).

To gain further insight into the regulatory mechanisms underpinning the CLL risk loci, we performed a quantitative trait locus (QTL) analysis using 487 genotyped CLL cases[14], with corresponding H3K27ac ChIP-seq profiles ($n = 97$)[12], chromatin accessibility data assessed by ATAC-seq ($n = 99$)[12], and 450k Illumina DNA methylation profiles ($n = 486$)[14] (Fig. 2a and Supplementary Data 6). Nine of the risk loci were typified by H3K27ac, 14 risk loci by chromatin accessibility, and 28 risk loci by DNA methylation QTLs (Fig. 2a, b and Supplementary Data 7). Within overlapping H3K27ac and accessibility QTLs (13 accessibility QTL peaks overlap with 10 of the H3K27ac QTL peaks, covering 5 risk loci), the direction of effect of associated epigenetic changes was concordant. Furthermore, overlapping DNA methylation QTLs within these regions showed opposite effects, i.e., higher DNA methylation with lower activity/accessibility (Supplementary Data 7). These data are consistent with risk alleles mediating their effects by influencing chromatin activity.

**Biological mechanisms and significance of CLL risk loci.** We next focused on the underlying biological mechanisms through which genetic variants at CLL risk loci shape the regulatory genome. Both our study (Fig. 1a) and published data[2,16,18] indicate that cancer GWAS risk variants map to regulatory regions, suggesting that germline genotypes may mediate their affect through altered transcription factor (TF) binding. In view of this, we assessed whether TF binding may be affected by genetic variation at chromatin-accessible regions of CLL risk loci. We used motifbreakR to predict disruption of TF-binding motifs and examined lymphoblastoid cell line (LCL) ChIP-seq data from the ENCODE project (Supplementary Data 8). This analysis revealed disruption of SPI1- and NFKB-binding sites at the risk alleles of SNPs rs13149699 ($r^2 = 0.81$ with sentinel SNP rs57214277) and rs539846 ($r^2 = 0.95$ with sentinel SNP rs8024033), respectively, as previously reported[2,19]. We also detected disruption of SPI1-binding sites at risk alleles of five other risk loci, associated with sentinel SNPs rs1002015, rs11637565, rs142215530, rs210143, and rs6489882, as well as disruption of other TFs relevant for B-cell development, including PAX5 and MEF2 TF family members (Fig. 3, Supplementary Data 8). Furthermore, we observed increased binding affinities of the TFs previously associated with de novo active regions in CLL, that is, FOX, NFAT, and TCF/LEF family members[12], at risk alleles of 16 risk loci, associated with sentinel SNPs rs11637565, rs4368253, and rs4869818, among others

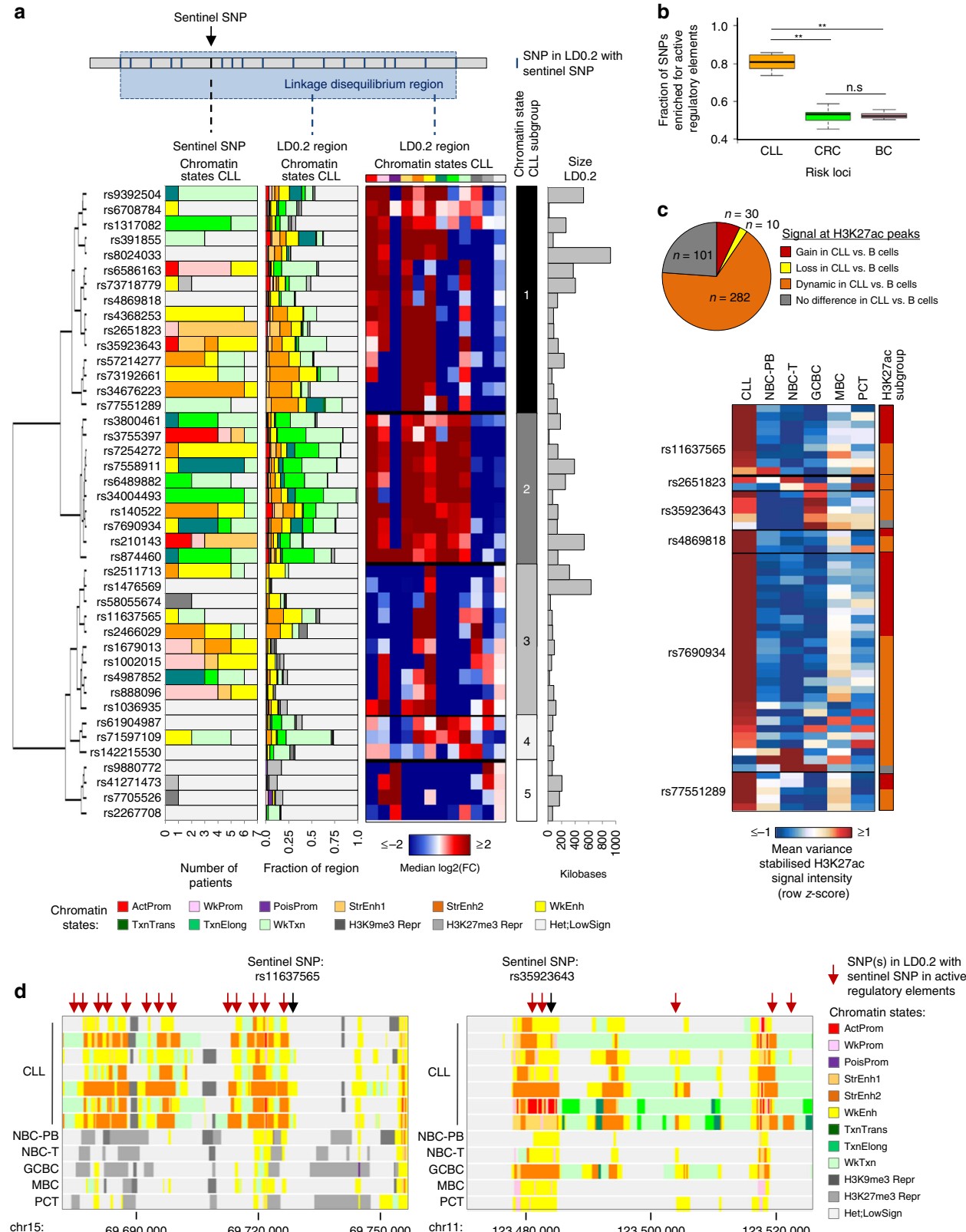

(Fig. 3, Supplementary Data 8). These findings underline the role of these TFs in CLL pathogenesis.

In addition, as risk variants may exert their pathogenic effect by altering the accessibility of affected alleles, we studied the existence of allelic imbalance at ATAC-seq peaks. The power to detect allelic imbalance in chromatin accessibility was restricted to sites with common variants and relatively large effects. Nevertheless, 10% of the risk loci (4/42) showed evidence of allelic imbalance for chromatin accessibility at a total of 9 SNPs (Supplementary Data 9). Eight of these SNPs (89%) overlapped with accessibility QTLs, a significantly larger fraction than for SNPs that did not show allelic imbalance (35/180, 19%, $P = 3.1 \times 10^{-5}$, Fisher's exact test).

**Fig. 1** Linking chronic lymphocytic leukemia (CLL) risk loci to chromatin states. **a** Top panel: graphical representation of a risk locus with a sentinel single-nucleotide polymorphism (SNP; representing the SNP with the strongest association in relation to CLL development) and its linkage disequilibrium (LD) region (shaded blue), containing all the SNPs that are within LD0.2 (LD with $r^2 \geq 0.2$) with the sentinel SNP. Lower panel: frequency of chromatin states in CLL ($n = 7$ biologically independent samples) at the exact position of the sentinel SNPs of the 42 CLL risk loci and in their LD regions (LD0.2 regions), as well as the median log2(fold change) of the different chromatin states in CLL in the LD0.2 regions. On the left, the sentinel SNPs are indicated, and on the right the different chromatin state subgroups (groups 1–5) of the risk loci, as referred to in the main text, as well as the size of the LD0.2 regions in kilobases. **b** Boxplots of fraction of CLL ($n = 42$ independent regions), colorectal (CRC, $n = 75$ independent regions), and breast cancer (BC, $n = 165$ regions independent regions) risk loci enriched for active regulatory elements in CLL cases ($n = 7$ biologically independent samples). Mean, median, minimum and maximum fraction of regions, and number of data points for CLL: 0.81, 0.81, 0.74, 0.86, 7; for CRC: 0.52, 0.53, 0.45, 0.59, 7; and for BC: 0.53, 0.52, 0.50, 0.56, 7. Corrected $P$ value, test statistics ($W$), median difference, and 95% confidence interval for CLL vs. CRC: $6.4 \times 10^{-3}$, 49, 0.28, 0.23–0.34, for CLL vs. BC: $6.4 \times 10^{-3}$, 49, 0.29, 0.23–0.33, and for CRC vs. BC: 1.0, 24, 0.01, −0.03 to 0.05. $P$ values were calculated using Wilcoxon rank-sum test (two-sided) and corrected using the Bonferroni multiple testing correction. **\*\*$P$-value $< 1.0 \times 10^{-2}$**, n.s. non significant. **c** Upper panel: Distribution of non-individual H3K27ac peaks with specific activation patterns in CLL as compared to normal B cells. Lower panel: Mean H3K27ac signal in CLL and normal B cells in all non-individual H3K27ac peaks of the selected risk loci. On the right, the different H3K27ac state subgroups are indicated, using the same color codes as the upper panel. Sample sizes were for CLL: $n = 7$ biologically independent samples and for NBC-PB, NBC-T, GCBC, MBC, and PC-T: $n = 3$ biologically independent samples. **d** Representation of chromatin states in seven CLL cases and normal B cells (one representative samples for each subpopulation) in the LD0.2 regions of the sentinel SNPs rs11637565 (left panel) and rs35923643 (right panel). The black arrows represent the sentinel SNPs, the red arrows indicate SNPs that are in LD0.2 with the sentinel SNP that are located in active regulatory elements (ActProm, StrEnh1, StrEn2) in at least one CLL case. Red arrows may point to clusters of SNPs in close proximity to each other. Genomic regions represented (GRCh38) are chr15:69,671,700–69,756,772 (left panel) and chr11:123,470,881–123,525,966 (right panel). ActProm active promoter, WkProm weak promoter, PoisProm poised promoter, StrEnh1 strong enhancer 1, StrEnh2 strong enhancer 2, WkEnh weak enhancer, Txn_Trans transcription transition, Txn_Elong transcription elongation, Wk_Txn weak transcription, H3K9me3_Repr H3K9me3 repressed, H3K27me3_Repr H3K27me3 repressed, Het;LowSign heterochromatin;low signal, NBC-PB naive B cell from peripheral blood, NBC-T naive B cell from tonsil, GCBC germinal center B cell, MBC memory B cell, PC-T plasma cell from tonsil

We finally sought to infer the biological relevance of risk loci using two complementary approaches. First, we performed a gene expression QTL (eQTL) analysis to identify candidate target genes using GeneChip Human Genome U219 gene expression data from 452 CLL cases[14]. Since genomic spatial proximity and long-range chromatin looping interactions are central to the regulation of gene expression, eQTL analysis was performed considering the expression levels of genes located within respective topologically associating domains (TADs)[20] spanning the LD regions (Supplementary Data 1). We detected 36 eQTLs (defined by 62 probe sets assigned to 37 target genes) associated with 23 risk loci (Fig. 2a, b and Supplementary Data 7). Included were many genes with established roles in CLL oncogenesis (e.g. *LEF1*), B-cell development (e.g. *IRF8*), or apoptosis (e.g. *BCL2L11*, *CASP8*, *FAS*, *BMF*). Of note, three genes highlighted by the eQTL analysis, *LEF1* (rs7690934), *IPCEF1* (rs4869818), and *DMRTA1* (rs1679013), have been previously identified as target genes of de novo active regions in CLL (Supplementary Data 5) and show higher expression in CLL compared with normal B cells[12]. Such observations suggest a role for these genes in both CLL predisposition and pathogenesis. Overall, 36 of the 42 risk loci (86%) showed a significant QTL in at least one of the four layers analyzed (Fig. 2b).

Second, we made use of in situ promoter capture Hi-C (CHi-C) data to examine chromatin looping interactions linking the LD-defined risk loci to promoters of candidate target genes. We analyzed CHi-C information from CLL cells as well as from naive and total B cells[12,21] (Supplementary Data 10). As well as confirming the interaction between the rs2466029-associated LD region and MYC at 8q24.21[22], the looping interactions implicate a number of other genes with established roles in B-cell development. Notably, these include *BCL2* as the target gene of rs77551289/rs4987852 variation at 18q21.33 and *BCL6* as the target gene of rs73192661 variation at 3q28. Overall, there was supportive evidence from the CHi-C data for chromatin interactions between the LD-defined CLL risk loci with promoters of 15 of the 37 candidate genes (41%) identified in the eQTL analysis, including *TLE3* as a target of the rs11637565 LD region as well as *UBR5* as a target of the rs2511713LD region (Fig. 4).

## Discussion

Characterization of the regulatory elements that are perturbed by CLL GWAS risk variants is central to efforts to define the mechanisms through which these variants operate and to reconstruct the biological networks that underlie CLL tumorigenesis. However, genome-wide epigenomic characterization of CLL in combination with the availability of genetic data in large CLL series has so far been lacking. Here we have addressed this deficit by using a large, genetically well-characterized CLL series[14] for which genome-wide data of multiple epigenetic layers were recently generated[12]. We combined the genetic and epigenetic data to determine in unprecedented detail genotype-dependent patterns of chromatin accessibility, activity, and DNA methylation in CLL. Overall, our approach has allowed us to infer potential functional variants at 14 of the CLL risk loci (Supplementary Data 11). Moreover, our analyses underline that risk loci in CLL affect genes that participate in interconnecting cellular pathways that are central to B-cell function, including immune response (SP140, BCL6, OAS1, and IRF8)[23–26], apoptosis (BCL2L11, CASP8, CFLAR, FAS, BMF, and BCL2)[27], and Wnt signaling (UBR5, TLE3, and LEF1)[28,29]. This information serves to illustrate the value of delineating the functional mechanisms and target genes underlying risk loci as such data can have an impact on the successful development of new therapeutic agents. In this respect, it is notable that the recently introduced CLL treatment Ibrutinib inhibits BTK[30], a kinase with a key role in B-cell function, while Venetoclax targets the antiapoptotic activity of BCL2[31]. In conclusion, our approach has allowed us to generate a refined regulatory map of CLL risk loci offering a more granular annotation of their functional impact, which will facilitate future experimental validation of potential causal variants.

## Methods

**Patients**. The clinical and biological characteristics of the 502 CLL patients studied are detailed in Supplementary Data 6. Cases were defined as IGHV-mutated when the identity of immunoglobulin genes was <98%. Tumor samples were obtained pre-therapy. All patients gave informed consent for their participation in the study following the International Cancer Genome Consortium (ICGC) guidelines and the ICGC Ethics and Policy committee[32], and this study was approved by the clinical research ethics committee of the Hospital Clinic of Barcelona.

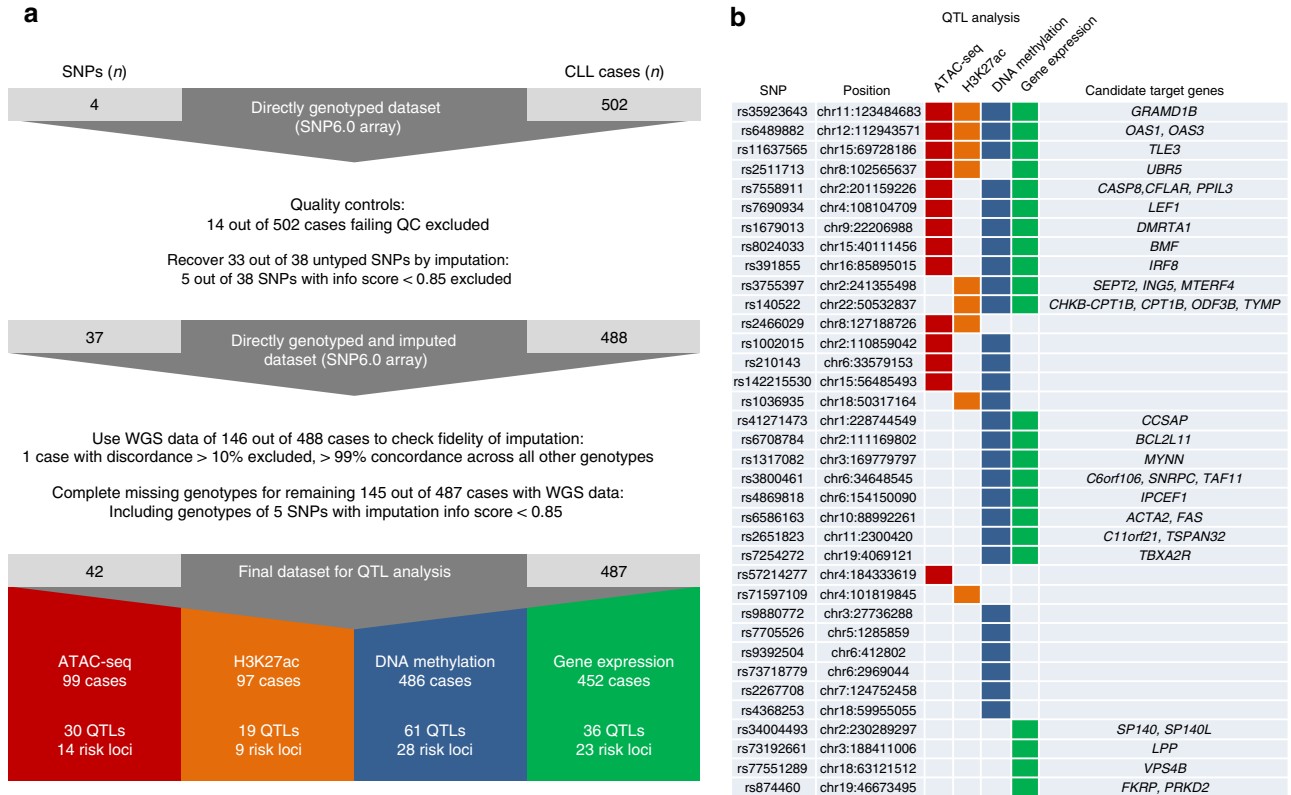

**Fig. 2** Quantitative trait locus (QTL) analysis and results overview. **a** Workflow describing the generation of genotypes for 42 non-HLA risk single-nucleotide polymorphisms (SNPs) in 487 chronic lymphocytic leukemia (CLL) patients for use in QTL analyses. The genotypes were assigned using a combination of directly genotyped ($n = 4$ independent SNPs) or imputed ($n = 38$ independent SNPs) SNP6.0 array data (502 independent CLL patients) and whole-genome sequencing (WGS) data ($n = 146$ independent CLL patients). Fourteen of the 502 CLL cases were excluded owing to non-European ancestry, excessive heterozygosity, or sample relatedness. Five of the 38 imputed SNPs were subject to poor imputation with an information measure <0.85. The imputation fidelity check using the WGS data resulted into the exclusion of one CLL case with >10% discordance between the imputed and WGS data. In the lower part of the panel, summary results for each type of analysis are shown. **b** Per locus overview of QTL analysis results. Loci are denoted by their respective sentinel SNPs (with their location in GRCh38) and shaded boxes indicate the presence of ≥1 significant QTL at false discovery rate <0.05 for each of the four layers analyzed: assay for transposase-accessible chromatin using sequencing (QTLs marked by dark red shading); H3K27ac (orange); DNA methylation (blue); gene expression (green). In addition, on the right, the candidate target genes identified by expression QTL analysis are indicated

**Datasets**. Whole-genome sequencing (WGS) and SNP6.0 array data (Thermo Fisher Scientific) of, respectively, 146 and 502 patients were obtained from Puente et al.[14]. Chromatin state segmentations (200 bp bins) of 7 CLLs and 15 normal B-cell samples as well as variance-stabilized (vst) data signals of H3K27ac and ATAC-seq determined by DESeq2 (corrected for signal proportion of tags (SPOT) score) (97 and 99 CLL patients, respectively) were derived from Beekman et al.[12] (http://resources.idibaps.org/paper/the-reference-epigenome-and-regulatory-chromatin-landscape-of-chronic-lymphocytic-leukemia). DNA methylation data of 486 CLLs (Illumina 450k arrays) and GC-RMA normalized gene expression data of 452 CLL patients (GeneChip Human Genome U219 arrays) were derived from Puente et al.[14].

**Definition of risk loci**. Sentinel SNPs at the 42 non-HLA risk CLL GWAS loci were obtained from Law et al.[2]. Sentinel SNPs at 169 BC risk loci were obtained from Michailidou et al.[16]. Sentinel SNPs at 78 CRC risk loci were obtained from Schmit et al.[15]. LD regions were defined by $r^2 \geq 0.2$ with each sentinel SNP. LD metrics were calculated using LDlink[33] for the European populations of the 1000 Genomes project (Phase 3 data). Three BC risk loci (rs554219, 4:84370124, rs373038216) were absent in the 1000 Genomes data and therefore excluded (LD regions could not be calculated). One BC risk locus (rs8176636) could not be mapped to the GRCh38 genome build and was therefore excluded. Three CRC risk loci were excluded (rs7758229, rs6061231, rs11064437) as they have only been reported in Asian populations.

**Linking risk loci to chromatin states**. Chromatin states at risk loci were assigned using the chromatin state segmentations as previously defined[12]. For each CLL

sample, the chromatin state enrichment at a CLL risk region, $E$, was calculated as

$$E_{ij} = \log_2 \frac{C_{j,i}}{C_{j,B}} \tag{1}$$

where $i$ is a CLL risk allele, $j$ is a chromatin state, $C_{j,i}$ is the fraction of the LD region ($r^2 \geq 0.2$) of $i$ covered by chromatin state $j$, and $C_{j,B}$ is the fraction of the background regions covered by chromatin state $j$. These background regions were determined as the merged LD regions ($r^2 \geq 0.2$) of all SNPs in the 1000 Genomes Project and UK10K[34], excluding regions containing CLL, BC, or CRC risk loci (Supplementary Data 12). The difference in chromatin state enrichment for CLL risk loci with BC and CRC risk loci was calculated using Wilcoxon rank-sum test. No covariates were tested. To account for multiple testing, a Bonferroni correction was applied.

**Comparison of H3K27ac signal among CLL and normal B cells**. Differential signal intensity of non-individual H3K27ac peaks was calculated using DESeq2[35]. H3K27ac counts at these loci were mined from Beekman et al.[12]. The analysis was performed by contrasting CLL with each of the normal subpopulations samples using nbinomWaldTest, correcting for the SPOT score; per sample, its condition (CLL or the corresponding normal B-cell subpopulation) and the SPOT score were introduced into the model as per Beekman et al.[12]. Regions with a false discovery rate (FDR) < 0.01 were considered significantly enriched.

**Assignments of genotypes**. Genotypes of (i) 42 sentinel SNPs, (ii) 225 bi-allelic SNPs located in ATAC-seq peaks analyzed in the QTL analysis being in LD with the sentinel SNP, and (iii) 94,233 bi-allelic background SNPs were assigned from

| Potential functional SNP (sentinel SNP) | Transcription factor | Risk allele | Binding affinity risk allele | Motif |
|---|---|---|---|---|
| rs210142 (rs210143) | SPI1 | C | Decreased | |
| rs12591150 (rs142215530) | PAX5 | T | Decreased | |
| rs4767033 (rs6489882) | MEF2A | T | Decreased | |
| rs919054 (rs11637565) | TCF3 | G | Increased | |
| rs7761411 (rs4869818) | NFATC1 | T | Increased | |
| rs8083367 (rs4368253) | FOXI1 | G | Increased | |

**Fig. 3** Transcription factor (TF)-binding motif analysis. Examples of altered TF-binding motifs at potential functional risk loci. The affected nucleotides in the different motifs are marked by an asterisk. The sequence surrounding the single-nucleotide polymorphism (SNP) is located under the motif graphs, with the different possible alleles of the potential functional SNP indicated below each other

SNP6.0 array data (using normal DNA) and WGS data (using normal and/or tumor DNA). The first set of peaks was used for QTL analyses, the latter two for the allelic imbalance analysis (see also Supplementary Data 9 and the paragraph on allelic imbalance in this "Methods" section). Genotyping data from the 502 CLL cases were subject to standard quality control[36], resulting in the exclusion of 14 cases prior to imputation because of non-European ancestry (using the HapMap version 2 CEU, JPT/CHB, and YRI populations as reference), excessive heterozygosity, or relatedness. SNPs with a call rate <95%, a minor allele frequency <1%, or displaying significant deviation from Hardy–Weinberg equilibrium (i.e., $P < 10^{-6}$) were also excluded. Whole-genome imputation was performed using the IMPUTE2 v2.3 software[37] with a merged reference panel consisting of data from the 1000 Genomes Project (phase 1 integrated release 3, March 2012) and UK10K[34]. No genotypes were assigned to poorly imputed SNPs with an information measure <0.85 (sentinel SNPs, $n = 5$; SNPs for allelic imbalance, $n = 31$; SNPs with low information measures were not included in the background set of 94,233 SNPs). No genotypes were assigned to SNPs residing in regions with copy number alterations in the corresponding CLL samples (as determined from the SNP6.0 array data of the matched tumor DNA). Imputation probabilities were converted to SNP genotypes using PLINK v1.9[38], using an uncertainty threshold of >0.1 to preclude genotype assignment. In order to check the fidelity of imputation and to look for potential mismatches in the CLL samples, we examined WGS data from 146 CLL cases for the sentinel SNPs (the cases overlapping with the 488 patients prior to imputation) and 38 CLL cases for the SNPs for the allelic imbalance analysis and the background SNPs (the cases overlapping with the 99 samples with ATAC-seq data). For the WGS data, SNPs were called with the MAQ consensus model implemented in SAMtools in those coordinates with at least 10 reads with a minimum mapping quality of 30 and base quality of 30[14]. No genotypes were assigned for the WGS data to SNPs showing discordant genotypes between normal and tumor DNA. One CLL sample (CLL618) where the genotypes assigned by imputation and WGS showed <90% concordance for the sentinel SNPs was excluded from all subsequent analyses. For the SNPs where genotypes assigned by imputation and WGS showed <90% concordance, we used WGS data only ($n = 0$ for sentinel SNPs and SNPs for allelic imbalance analysis; $n = 563$ for the 94,233 background SNPs). Upon discordance between the genotypes of the imputation and the WGS data for the remaining SNPs and samples, we used the WGS assigned genotypes. We also used WGS to add missing genotypes that had failed to be assigned following imputation (SNP information measure <0.85 or genotype probability <0.90), generating matrices of 42 sentinel SNPs genotyped in up to 487

cases for QTL analyses (Fig. 2a) and of 225 SNPs as well as 94233 background SNPs genotyped in up to 99 cases for allelic imbalance analysis.

**QTL analysis**. QTL analyses were performed using genotype data for the sentinel SNPs in conjunction with ATAC-seq ($n = 99$)[12] H3K27ac ($n = 97$)[12], methylation ($n = 486$)[14], and gene expression data ($n = 452$)[14] from CLL cases. Input data were vst-normalized data signals (corrected for SPOT score) for ATAC-seq and H3K27ac, $M$-values for DNA methylation, and GC-RMA-normalized for gene expression. ATAC-seq and H3K27ac QTL analyses were performed for peaks present in at least 10% (with a minimum of two) of the patients in one or more of the following subgroups: homozygous non-risk, heterozygous risk, or homozygous risk based on the sentinel SNP genotype. eQTL analysis was restricted to expressed probes (GC-RMA levels >4.5) again in at least 10% (with a minimum of two) of the patients in one or more of the following subgroups: homozygous non-risk, heterozygous risk, or homozygous risk based on the sentinel SNP genotype. To account for confounding factors, we implemented the probabilistic estimation of expression residuals (PEER) method[39] using 10, 10, 45, and 65 factors for ATAC-seq, H3K27ac, methylation, and gene expression data, respectively. Inverse normal-transformed PEER-computed residuals were used as input for QTL analyses, which were conducted using Matrix eQTL, under a linear model[40]. TADs for the eQTL analysis were defined using in situ Hi-C data from the LCL GM12878[41] and were mined from Beekman et al.[12] (http://resources.idibaps.org/paper/the-reference-epigenome-and-regulatory-chromatin-landscape-of-chronic-lymphocytic-leukemia).

**Promoter CHi-C**. Raw data of previously generated in situ promoter capture Hi-C (CHi-C) on CLL, naive, and total B cells was used[12,21]. Reads were aligned to the GRCh37 build using bowtie2 v2.2.6[42] and identification of valid di-tags was performed using HiCUP v0.5.9[43]. To declare significant contacts, HiCUP output was processed using CHiCAGO v1.1.8[44]. As advocated, interactions with a score ≥5.0 were considered to be statistically significant.

**Allelic imbalance**. A beta-binomial test was used to assess the allelic imbalance at all bi-allelic SNPs that (i) were in LD ($r^2 \geq 0.2$) with the sentinel SNP and located in ATAC-seq peaks used for the QTL analysis ($n = 225$), and (ii) had at least 3 heterozygous samples and at least 10 reads, leading to a final number of 189 SNPs

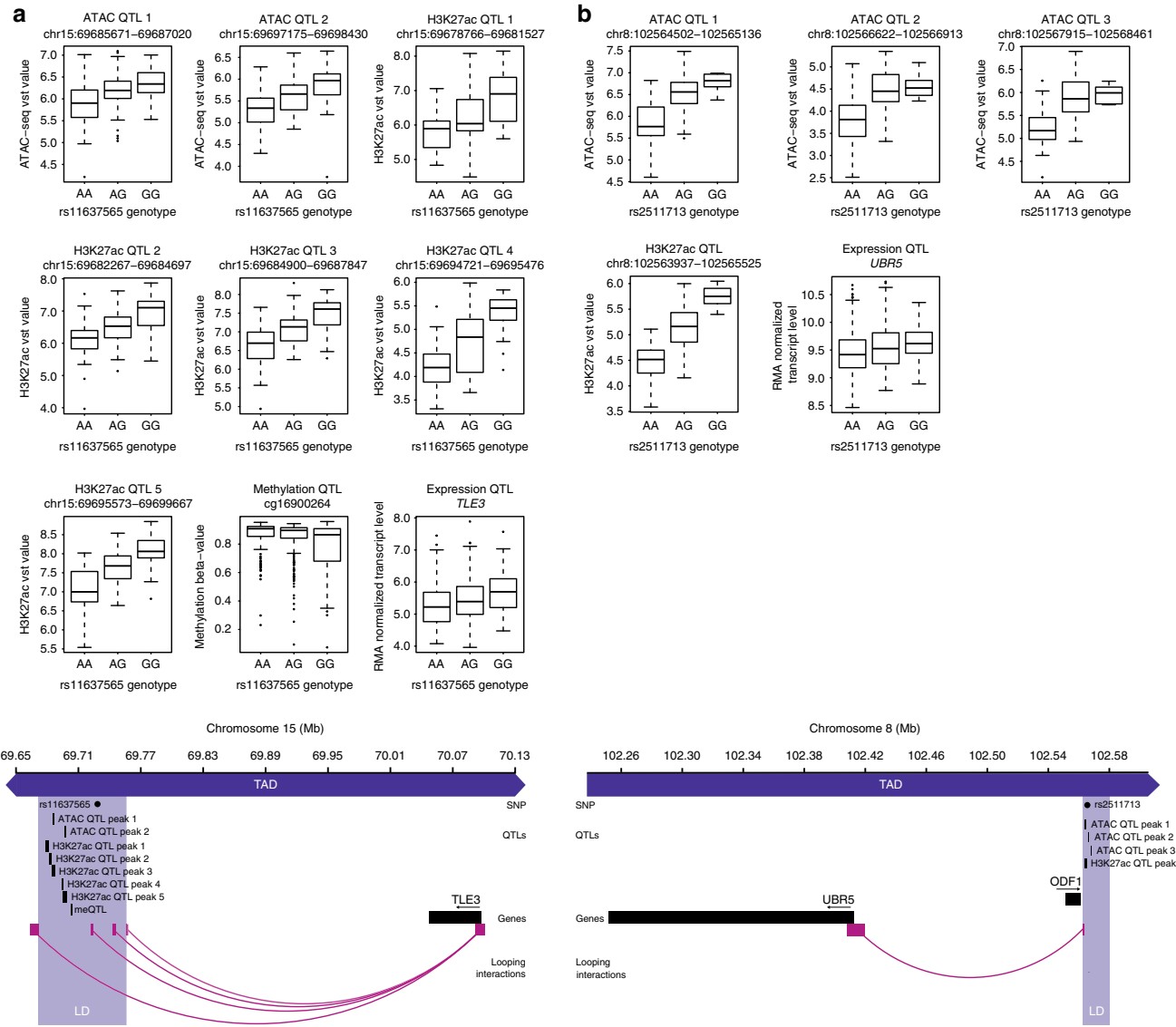

**Fig. 4** Characterization of quantitative trait loci (QTLs) and looping interactions at the 15q23 and 8q22.3 chronic lymphocytic leukemia (CLL) risk loci. (Top panels) Boxplots illustrating associations between **a** rs11637565 and **b** rs2511713 genotypes and chromatin accessibility (assay for transposase-accessible chromatin(ATAC) QTL), histone H3 lysine 27 acetylation (H3K27ac QTL), DNA methylation (methylation QTL), and candidate target gene transcript levels (expression QTL) in CLL patients (number of independent CLL samples for ATAC: $n = 99$, H3K27ac: $n = 97$, DNA methylation: $n = 486$, and transcript levels: $n = 452$). (Lower panels) Chromosomal positions of sentinel single-nucleotide polymorphisms (SNPs; rs11637565/rs2511713, gray dots), QTL peaks/probes, and coding genes (black rectangles). The linkage disequilibrium (LD) regions, encompassing SNPs in LD $r^2 \geq 0.2$ with the sentinel SNPs are indicated by light blue rectangles. Also shown are looping interactions (pink arcs) from the LD regions to the promoters of TLE3 and UBR5. Pink rectangles represent interacting HindIII fragments. Topologically associating domains (dark blue rectangles, TAD) containing all features are truncated for clarity; their extension beyond the figure boundaries is indicated by arrowheads

to be assessed. Reads with mapping bias were removed using WASP[45] and bwa 0.7.15[46]. In brief, the nucleotide sequence of the reads overlapping the investigated SNPs was modified to investigate potential mapping bias. More specifically, the nucleotides at known SNP positions within these reads were substituted by nucleotides of the other possible allele(s). Reads that after modification failed to remap in the same position in the genome were discarded. PCR duplicates were removed using the WASP script rmdup_pe.py. Thereafter, for each analyzed SNP the number of reads containing the two different possible alleles was determined, pooling all heterozygote samples. To control for possible errors in genotyping, counts from samples at SNPs were only used if they presented a reference allele count ratio of 0.1–0.9 in comparison to the total counts of the reference and the alternative allele together. The beta-binomial parameters were estimated using bi-allelic SNPs not in LD with the sentinel SNP that were located (i) within ATAC-seq peaks used for the QTL analysis and (ii) within genome-wide ATAC-seq peaks harboring a peak in at least 10% of the CLL patients (94233 SNPs in total, SNPs within the HLA locus, the immunoglobulin loci, and imprinted regions were excluded). The maximum likelihood estimations were $\alpha = 20.5$ and $\beta = 20.3$. The P

values were corrected for multiple testing (FDR < 0.1). The central tendency of allelic imbalance was defined per SNP by calculating the allele ratio of the reference allele, while its 95% confidence intervals was calculated using the normal approximation.

**TF motif and binding analysis**. To investigate TF-binding site disruption, the motifbreakR package[47] was used. Scores were calculated using the relative entropy algorithm. P values were calculated using the score, given the score distribution using the position weight matrices. SNPs were also annotated for bound TFs using ENCODE[48] ChIP-seq data for LCLs.

**GWAS datasets**. Meta-analysis of three GWAS datasets (UK-CLL1, UK-CLL2 and the NHL-GWAS Consortium)[2,3] was conducted as previously described[2,3]. NHL-GWAS Consortium data were a subset of those previously published[2,7,8] and were downloaded from dbGAP phs000801.v2.p1 (sub-study phs000802.v2.p1). After standard QC procedures, final GWAS datasets comprised of UK-CLL1:

503 CLL cases and 2698 controls; UK-CLL2: 1304 CLL cases and 2501 controls; and NHL-GWAS: 1751 cases and 2494 controls. Imputation of untyped SNPs in the GWAS datasets was performed using IMPUTE2 v2.3 software and a merged reference panel consisting of data from 1000 Genomes Project (phase 1 integrated release 3 March 2012) and the UK10K, as previously described[2,3]. Tests of association between imputed SNPs and CLL were performed using logistic regression under an additive genetic model in SNPTESTv2.5[49]. Meta-analysis was performed using the fixed-effects inverse-variance method based on the $\beta$ estimates and standard errors from each study using META v1.6[50].

**Reporting summary**. Further information on research design is available in the Nature Research Reporting Summary linked to this article.

## Data availability

All raw data for this study were mined from previous studies[12,14,21] and has been deposited at the European Genome-Phenome Archive (EGA; http://www.ebi.ac.uk/ega/), which is hosted at the European Bioinformatics Institute (EBI), under accession numbers EGAS00000000092, EGAD00001004046, EGAS00001000272, and EGAS00001001911. Furthermore, processed data matrices can be found in http://resources.idibaps.org/paper/insight-into-genetic-predisposition-to-chronic-lymphocytic-leukemia-from-integrative-epigenomics.

## Code availability

Custom code related to the present article can be found in http://resources.idibaps.org/paper/insight-into-genetic-predisposition-to-chronic-lymphocytic-leukemia-from-integrative-epigenomics.

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

## Acknowledgements

This work was funded by the European Union's Seventh Framework Programme through the Blueprint Consortium (grant agreement 282510), the International Cancer Genome Consortium (Chronic Lymphocytic Leukemia Genome consortium to E.C. and C.L.-O.), La Caixa Foundation (CLLEvolution-HE17–00221, to E.C. and C.L.-O.), the World Wide Cancer Research Foundation Grant No. 16–1285 (to J.I.M.-S.), the Spanish Ministerio de Economía y Competitividad (MINECO), Grant No. SAF2015-64885-R (to E.C.) and Grant No. PMP15/00007, part of Plan Nacional de I+D+I and co-financed by the ISCIII-Sub-Directorate General for Evaluation and the European Regional Development Fund (FEDER-Una manera de Hacer Europa) (to E.C.), the Generalitat de Catalunya Suport Grups de Recerca AGAUR 2017-SGR-736 (to J.I.M.-S.) and 2017-SGR-1142 (to E.C.), the CERCA Programme/Generalitat de Catalunya, and CIBERONC. In the UK, funding was provided by Bloodwise (LRF05001, LRF06002, and LRF13044) with additional support from Cancer Research UK, the Arbib Fundand the Thomas Roberts Trust. H.E.S. was supported by a fellowship from the Lady TATA Memorial Trust (International Award). R.B. was supported by fellowships from the EU (Marie Skłodowska-Curie Inter European Fellowship) and the Lady TATA Memorial Trust (International Award) and has received financial support through the Postdoctoral Junior Leader Fellowship Programme from the La Caixa Banking Foundation. E.C. is an Academia Researcher of the Institució Catalana de Recerca i Estudis Avançats (ICREA) of the Generalitat de Catalunya. This work was partially developed at the Centro Esther Koplowitz (CEK, Barcelona, Spain).

## Author contributions

Investigator contributions were as follows: H.E.S., R.B., V.C., G.O., P.J.L., G.C., and M.P. contributed to data analysis; H.E.S., R.B., P.J.L., D.M.-G., J.G.-A., S.B., and X.S.P. contributed to genotype assignments; V.C., P.J.L., C.L.-O., D.C., D.T, J.M.A., and E.C. participated in the study design and data interpretation together with H.E.S., R.B., R.S.H., and J.I.M.-S.; H.E.S., R.B., R.S.H., and J.I.M-S. directed the research and wrote the manuscript.

## Additional information

**Competing interests:** The authors declare no competing interests.

