## [Peer Review File · Nature Communications]

Reviewers' comments:

Reviewer #1 (Remarks to the Author):

The authors present a meta-analysis that combines two recently published datasets, namely GWAS data and epigenome profiling for CLL. As expected, some of the GWAS hits lie genomic regions that are characterized by open chromatin in CLL cells (*). Moreover, speculation about the functional role of specific target genes are presented but not validated.

This short manuscript stays entirely descriptive and does not provide a substantial advance in our understanding of CLL nor any relevant resource of methodological advance over previously published research by the same authors (and others).

To make the study suitable for Nature Communications (or any major journal), the authors should use their data to select 2-3 interesting regulatory SNPs (roughly equivalent of Figure 1 of a convincing paper), do fine mapping as needed and then perform comprehensive functional / experimental workup to understand the mechanistic role in CLL for the selected SNPs (roughly equivalent of what would be Figure 2 to 5 in a convincing paper).

(*) While it has been reported in many studies (and is therefore plausible to assume here as well) that GWAS hits are significantly enriched for cell type specific regulatory regions, this does not obviate the need for rigorous statistical analysis in the current manuscript. Unfortunately, the authors fail to control sufficiently well for the uneven distribution of regulatory regions across the genome, such that even a set of GWAS hits for e.g. breast or colon cancer would likely show significant enrichment for the regulatory regions that the authors identify in CLL cells. Moreover, the authors in several instances fail to provide a measure of the effect size for specific enrichments tested.

Reviewer #2 (Remarks to the Author):

The investigation of regulatory mechanisms at chronic lymphocytic leukemia (CLL) loci is important. GWAS studies have provided evidence for CLL genetic predisposition but the definition of which mechanisms are actually involved involved has been constrained by lack of integrated genomic data especially in large series.

The here presented study on the detailed epigenomic characterization of CLL samples as well as of different normal B-cell subsets and the subsequent integration of genomic data represents a further step that advances the field and accelerates the identification of candidate mechanisms of CLL predisposition. This study presents an accurate refinement and provides the starting point for future experimental validation which hopefully will help understanding the biological networks that are central to CLL genesis

Reviewer #3 (Remarks to the Author):

In this study the authors perform detailed epigenomic analysis of 42 non-HLA loci which show strong association with increased risk of CLL. In the first part of the study, the authors compare H3K27ac, H3K4me3 and H3K4me1 ChIP-sequencing profiles for 42 LD regions for risk loci, between CLL patients and B-cells during different stages of development. Using this analysis, the authors demonstrate that these 42 LD regions show differential regulation during B-cell development. To further elucidate the impact of these SNPs of gene-regulation the authors perform a Quantitative trait analysis (QTL) analysis for chromatin accessibility (ATACseq), H3K27ac ChIPseq, DNA methylation and gene expression for these regions in large cohort of CLL

patients. In this analysis, the authors rigorously control for the impact of covariates and other potential sources of variations and report expected associations demonstrating the robustness of the results. For e.g. DNA methylation QTLs were negatively correlated with chromatin accessibility. Finally, the authors perform motif analysis and demonstrate with examples that SNPs such as rs13149699 and rs539846 disrupt binding sites for transcription factors relevant to B-cell development. While the methods used in this paper is robust and the dataset generated is unique, I have several concerns with the results, which I describe below:

1. The authors characterize CLL GWAS hits, by integrating them with chromatin data, and performing QTL analyses. This is nice, however, the subsequent follow-up and interpretation of the results is limited. For example, it would be nice if additional genes or pathways could be implicated by or if functional follow-up or experimental validation of some of the findings could be included. In summary the paper would be more impactful if the biological consequences of the SNPs/QTLs that they describe could be highlighted.

2. The use of chromatin states to characterize the LD0.2 regions surrounding sentinel SNPs is reasonable, but it is unusual to use them for quantitative comparisons between loci and cell types. For example, on lines 177-179 the authors state that a "difference in active chromatin... was greater than 10% we considered it to be significant". It would be better to perform comparisons of the individual chromatin marks for analyses like these and for assessing statistical significance. For example, significant differences in ChIP-seq peaks between conditions can be assessed using methods such as DE-seq or limma. Related to this, the statement in line 88 that "CLL risk loci are enriched for differential regulation in B-cell development" is a pretty broad claim, given that the authors don't perform a formal statistical test for an enrichment.

3. One concern with the allelic imbalance analysis is that the authors use a Binomial test. It has been demonstrated in several studies that allele-specific reads from next-generation sequencing experiments are overdispersed and p-values from binomial and related tests are typically deflated (e.g. Harvey, Gregory A Moyerbrailean et al. 2014; van de Geijn et al. 2015; Love et al. 2014). It would be better if the authors could use a Beta-Binomial test instead. Note that estimating the dispersion parameter for a Beta-Binomial test might require using data from many loci (not just those in the LD0.2 regions). It would also be useful if the the authors should include Quantile-Quantile plots to demonstrate that their allelic-imbalance tests are well calibrated.

Minor points:

4. It would be nice to include specific examples/figures of ATAC-seq and DNA-methylation QTLs which show differences in signal associated with genotype.

Reviewer 1

The authors present a meta-analysis that combines two recently published datasets, namely GWAS data and epigenome profiling for CLL. As expected, some of the GWAS hits lie genomic regions that are characterized by open chromatin in CLL cells (*). Moreover, speculation about the functional role of specific target genes are presented but not validated.

1. This short manuscript stays entirely descriptive and does not provide a substantial advance in our understanding of CLL nor any relevant resource of methodological advance over previously published research by the same authors (and others). To make the study suitable for Nature Communications (or any major journal), the authors should use their data to select 2-3 interesting regulatory SNPs (roughly equivalent of Figure 1 of a convincing paper), do fine mapping as needed and then perform comprehensive functional / experimental workup to understand the mechanistic role in CLL for the selected SNPs (roughly equivalent of what would be Figure 2 to 5 in a convincing paper).

Answer:

Although we agree with the reviewer that a thorough functional/experimental analysis is necessary to decipher the causal relationship between risk loci and CLL development, we hope that the reviewer understands that our study is actually a necessary intermediate step, and as such, has an important resource value for the community. We believe that our manuscript is the essential missing link between previously published research and future functional studies, as it fine tunes the genomic regions where potential functional risk variants are likely to be found. The methodological advance of our work lies in the fact that we provide new insights in the regulatory mechanisms at CLL risk loci by linking genetic and epigenetic profiles in up to 486 CLL primary cases while in previous series, as far as we are aware, cell line models and/or normal tissues were used to identify QTLs. Furthermore, to our knowledge this detailed multi-omics analysis is one of the first of its kind in the context of genetic predisposition and cancer.

Changes to the manuscript:

No changes were made to the manuscript

2. (*) While it has been reported in many studies (and is therefore plausible to assume here as well) that GWAS hits are significantly enriched for cell type specific regulatory regions, this does not obviate the need for rigorous statistical analysis in the current manuscript. Unfortunately, the authors fail to control sufficiently well for the uneven distribution of regulatory regions across the genome, such that even a set of GWAS hits for e.g. breast or colon cancer would likely show significant enrichment for the regulatory regions that the authors identify in CLL cells.

Answer:

We appreciate this comment. This is definitely an important issue that we did not address in the submitted version of the manuscript. In the review process, we have compared the enrichment of regulatory elements in CLL at CLL risk loci with that of breast cancer (BC) and colorectal cancer (CRC) risk loci. By doing this, we found a significant increase of active regulatory elements in CLL at CLL risk loci in comparison to BC and CRC risk loci. This finding suggests that risk loci associated with any cancer type are not equally located in regulatory elements in CLL samples, suggesting thus that CLL-specific risk loci are significantly biased towards CLL-specific regulatory elements.

Changes to the manuscript:

- Results section: The following sentence was added in the first paragraph of “The active chromatin landscape at CLL risk loci” section (page 4 of the revised manuscript without track changes): *“Specifically, on average 81% (34/42) of the LD-defined CLL risk loci showed enrichment of active promoters and/or enhancers per CLL sample (Fig. 1b, Supplementary Table 3). This enrichment of CLL-related active regulatory elements at CLL risk loci contrasts with the significantly lower enrichment observed at both colorectal and breast cancer GWAS loci (Fig. 1b, Supplementary Table 3).”*
- Method section: A description of the breast cancer and colorectal cancer loci was added to the section “Definition of risk loci” (page 7 of the revised manuscript without track changes).
- Figures: Figure 1b was added.
- Supplement: Supplementary table 3 was added.

3. Moreover, the authors in several instances fail to provide a measure of the effect size for specific enrichments tested.

Answer:

We agree with the reviewer that the effect sizes are important to provide to interpret the effects that we describe and that they are in several instances omitted in the manuscript. Therefore, we provide the effect size for all analyses provided in the manuscript in the supplementary tables, i.e. the enrichment of active regulatory elements in CLL, CRC and BC loci (newly added in the revision process), the QTL analyses (newly added in the revision process), the allelic imbalance (already available in the initial version of the manuscript) and the binding affinity of the TF analysis (already available in the initial version of the manuscript).

Changes to the manuscript:

- Changes to the supplement:
 - Supplementary 3 was added to show the median log₂ FC in chromatin state enrichment in primary CLL cases at all risk loci analysed.
 - Supplementary table 7 of the revised manuscript contains a column showing the effect sizes of the QTLs. DNA methylation data in this table was changed to mean to measure effect size.

Reviewer 2

The investigation of regulatory mechanisms at chronic lymphocytic leukemia (CLL) loci is important. GWAS studies have provided evidence for CLL genetic predisposition but the definition of which mechanisms are actually involved involved has been constrained by lack of integrated genomic data especially in large series.

The here presented study on the detailed epigenomic characterization of CLL samples as well as of different normal B-cell subsets and the subsequent integration of genomic data represents a further step that advances the field and accelerates the identification of candidate mechanisms of CLL predisposition. This study presents an accurate refinement and provides the starting point for future experimental validation which hopefully will help understanding the biological networks that are central to CLL genesis

Answer:

We thank this reviewer for the supportive comments and positive evaluation of our manuscript.

Changes to the manuscript:

No changes required.

Reviewer 3

In this study the authors perform detailed epigenomic analysis of 42 non-HLA loci which show strong association with increased risk of CLL. In the first part of the study, the authors compare H3K27ac, H3K4me3 and H3K4me1 ChIP-sequencing profiles for 42 LD regions for risk loci, between CLL patients and B-cells during different stages of development. Using this analysis, the authors demonstrate that these 42 LD regions show differential regulation during B-cell development. To further elucidate the impact of these SNPs of gene-regulation the authors perform a Quantitative trait analysis (QTL) analysis for chromatin accessibility (ATACseq), H3K27ac ChIPseq, DNA methylation and gene expression for these regions in large cohort of CLL patients. In this analysis, the authors rigorously control for the impact of covariates and other potential sources of variations and report expected associations demonstrating the robustness of the results. For e.g. DNA methylation QTLs were negatively correlated with chromatin accessibility. Finally, the authors perform motif analysis and demonstrate with examples that SNPs such as rs13149699 and rs539846 disrupt binding sites for transcription factors relevant to B-cell development. While the methods used in this paper is robust and the dataset generated is unique, I have several concerns with the results, which I describe below:

1. The authors characterize CLL GWAS hits, by integrating them with chromatin data, and performing QTL analyses. This is nice, however, the subsequent follow-up and interpretation of the results is limited. For example, it would be nice if additional genes or pathways could be implicated by or if functional follow-up or experimental validation of some of the findings could be included. In summary the paper would be more impactful if the biological consequences of the SNPs/QTLs that they describe could be highlighted.

Answer:

We agree with the reviewer that it is important to state the biological consequences of the results that we provide in the manuscript, and that this was not sufficiently addressed in the submitted version. We have now addressed this issue in two complementary ways. First of all, we have linked the LD regions with potential target genes including data on the three-dimensional (3D) chromatin structure in CLL and normal B cells. Secondly, we have provided a detailed description of the pathways that are targeted by the genes identified in the 3D chromatin and eQTL analysis.

Changes to the manuscript:

➤ Result section:

- The following sentence was included to the description of the eQTL analysis (page 5 of the revised manuscript without track changes): *“Included were many genes with established roles in CLL oncogenesis (e.g. LEF1), B-cell development (e.g. IRF8) or apoptosis (e.g. BCL2L11, CASP8, FAS, BMF).”*
- A final paragraph has been added to the result section, describing the 3D interactions determined by promoter capture Hi-C (Chi-C) (page 5-6 of the revised manuscript without track changes).

➤ Discussion: The following sentence was added (page 6 of the revised manuscript without track changes): *“Moreover, our analyses underline that risk loci in CLL affect genes that participate in*

interconnecting cellular pathways that are central to B-cell function, including immune response (SP140, BCL6, OAS1 and IRF8), apoptosis (BCL2L11, CASP8, CFLAR, FAS, BMF and BCL2) and Wnt signalling (UBR5, TLE3 and LEF1). This information serves to illustrate the value of delineating the functional mechanisms and target genes underlying risk loci as such data can have an impact on the successful development of new therapeutic agents. In this respect it is notable that the recently introduced CLL treatment Ibrutinib inhibits BTK, a kinase with a key role in B-cell function, whilst Venetoclax targets the anti-apoptotic activity of BCL2.”

- Method section: A description of the promoter capture Hi-C (CHi-C) data analysis was added to the method section (page 9 of the revised manuscript without track changes).
- Supplement: Supplementary table 10 was added, summarising the promoter capture Hi-C (CHi-C) results.

2. The use of chromatin states to characterize the LD0.2 regions surrounding sentinel SNPs is reasonable, but it is unusual to use them for quantitative comparisons between loci and cell types. For example, on lines 177-179 the authors state that a “difference in active chromatin... was greater than 10% we considered it to be significant”. It would be better to perform comparisons of the individual chromatin marks for analyses like these and for assessing statistical significance. For example, significant differences in ChIP-seq peaks between conditions can be assessed using methods such as DE-seq or limma. Related to this, the statement in line 88 that “CLL risk loci are enriched for differential regulation in B-cell development” is a pretty broad claim, given that the authors don’t perform a formal statistical test for an enrichment.

Answer:

Thank you for this detailed technical criticism, which highlights the need of providing statistical significance to the quantitative comparison between risk loci. Furthermore, we realised that our comparisons did not take into account that the activity state of different regulatory elements in one single LD region can be modulated in CLL and normal B cells in different ways. To address these issues, we have compared the H3K27ac signal (representing genomic activity) at 423 non-individual H3K27ac peaks located within the LD regions of the 42 risk loci using DE-seq2. We show that these peaks are significantly enriched for regions with increased activity in CLL as compared to all normal B cells and for regions that show dynamic H3K27ac levels among CLL and B cells.

Changes to the manuscript:

- Results section: At page 4 of the revised version of the manuscript without track changes, the old section stating:

“Since epigenetic regulation is closely aligned to B-cell development, we compared epigenetic profiles for each LD region in naive, germinal centre, and memory B-cells, plasma cells as well as CLL (Fig. 1b). Twenty-three regions showed a similar activation level in CLL and normal B-cells. The other regions showed differential activity across CLL and B-cells at different stages of differentiation (Fig. 1b). Furthermore, five of the 42 risk loci contained recently identified de novo active regions in CLL (Supplementary Table 3), including the loci marked by rs4869818, rs1679013 and rs4368253, which overlap with small (<2.7kb) de novo active regions in CLL (Supplementary Table 3).”

was replaced for:

“Since the epigenome of B cells varies according to its maturation state, we examined H3K27ac profiles at the risk loci in CLL and a range of normal B-cell subpopulations (Fig. 1c, Supplementary Table 4). H3K27ac signals at 10 risk loci (represented by 30 H3K27ac peaks) were significantly increased in CLL cells as compared to normal B-cell subpopulations, a significant higher number than expected ($P = 3.5 \times 10^{-3}$, Fisher’s exact test, 11 peaks based on 1000 permutations). These included five regions of de novo activation previously documented in CLL12 (Supplementary Table 5). Overall, 93% of the LD-defined risk loci (39/42, represented by 282 H3K27ac peaks) displayed differential H3K27ac profiles across CLL and the different normal B-cell subpopulations, also significantly more than that expected ($P = 3.0 \times 10^{-3}$, Fisher’s exact test, 239 peaks based on 1000 permutations). Collectively, our data show that CLL risk loci are enriched for CLL-related regulatory elements being CLL-specific or having differential regulation across CLL and B-cell development, as exemplified by the loci marked by the SNPs rs11637565 and rs35923643 (Fig. 1d)”.

- Method section: The section “Linking risk loci to chromatin states” was changed and the section “Comparison of H3K27ac signal among CLL and normal B cells” was added (page 7-8 of the revised manuscript without track changes).
- Figures: The old figure 1b was substituted for new figure 1c.
- Changes in the Supplement: Supplementary table 4 was added to the manuscript.

3. One concern with the allelic imbalance analysis is that the authors use a Binomial test. It has been demonstrated in several studies that allele-specific reads from next-generation sequencing experiments are overdispersed and p-values from binomial and related tests are typically deflated (e.g. Harvey, Gregory A Moyerbrailean et al. 2014; van de Geijn et al. 2015; Love et al. 2014). It would be better if the authors could use a Beta-Binomial test instead. Note that estimating the dispersion parameter for a Beta-Binomial test might require using data from many loci (not just those in the LD0.2 regions). It would also be useful if the the authors should include Quantile-Quantile plots to demonstrate that their allelic-imbalance tests are well calibrated.

Answer:

We thank the reviewer for pointing to this critical issue in our manuscript, and we would like to state that we have spent a large fraction of the time of the review process dealing with this issue. We have generated Quantile-Quantile plots for the p-values produced in the binomial test and we agree with the reviewer that the estimations of this test result in potential false positives in our 225 target SNPs (see Fig. R1 below, for reviewer’s perusal only). Therefore, as suggested by the reviewer we have defined a background set of 94233 bi-allelic SNPs. This background set contained SNPs that were not in LD with the sentinel SNP and that were located (i) within ATAC-seq peaks used for the QTL analysis and (ii) within genome-wide ATAC-seq peaks harbouring a peak in at least 10% of the CLL patients. Mapping bias was corrected for by WASP and genotyping errors were controlled by using only counts from samples at SNPs if they presented a reference allele count ratio of 0.1-0.9 in comparison to the total counts of the reference and the alternative allele together. Next, we obtained the parameters of the beta-distribution from the background set of 94233 SNPs using a maximum likelihood estimation, which were: $\alpha=20.5$ and $\beta=20.3$. By plotting histograms of the p-values produced in the beta-binomial test we observed a peak of p-values around zero (see Fig. R1 below, for reviewer’s perusal only), showing that a number of our background SNPs are affected by

allelic imbalance. We performed multiple steps to address this effect. In particular, we removed SNPs within the HLA locus, the immunoglobulin loci and imprinted regions as well as SNPs affected by copy number variations. In spite of these efforts, the problem was not completely corrected. Thus, the presence of these SNPs showing unexplained allelic imbalance in our background set led to a high estimation of over dispersion and as a consequence only few of the 225 target SNPs could be assigned as showing significant allelic imbalance as compared to the background. This implies that with this new approach, our results are most likely true positives, but at the same time, we most likely have false negatives. In spite of this caveat, we believe that it is more important to report few, but credible risk SNPs showing allelic imbalance. Therefore, we have decided to use the results derived from the beta-binominal test in the new version of the manuscript rather than keeping the previous results with the binomial test.

Figure R1. (Left panel) Quantile-quantile plot of p-values of the 225 target SNPs calculated using the binomial test. (Middle panel). Quantile-quantile plot of p-values of the 94233 background SNPs calculated using the binomial test. (Right panel) Histogram of p-values of the 94233 background SNPs of the beta-binomial test after performing the corrections explained above.

Changes to the manuscript:

➤ Results section:

- At page 5 of the revised version of the manuscript without track changes, the paragraph describing the allelic imbalance was modified from:

“Germline genotypes are an important determinant of allelic gene expression in tumors, partially as a consequence of altered transcription factor (TF) binding. Both our study (Fig. 1a) and published evidence indicate that cancer GWAS risk variants map to regulatory regions, suggesting that allelic imbalance at ATAC-seq peaks may exist (Fig. 3a). Although the power to detect allelic imbalance in chromatin accessibility was restricted to sites with common variants and relatively large effects, twenty-two risk loci showed evidence of allelic imbalance for chromatin accessibility at a total of 47 SNPs (Supplementary Table 6). Twenty-three of these SNPs (49%) overlapped with accessibility QTLs, a significantly larger fraction than for SNPs that did not show allelic imbalance (20/152, 13%, $P = 1.0 \times 10^{-6}$, Fisher’s exact test).”

to:

“In addition, as risk variants may exert their pathogenic effect by altering the accessibility of affected alleles, we studied the existence of allelic imbalance at ATAC-seq peaks. The power to detect allelic imbalance in chromatin accessibility was restricted to sites with common variants and relatively large effects. Nevertheless, 10% of the risk loci (4/42) showed evidence of allelic imbalance for chromatin accessibility at a total of 9 SNPs (Supplementary Table 9). Eight of these SNPs (89%) overlapped with accessibility QTLs, a significantly larger fraction than for SNPs that did not show allelic imbalance (35/180, 19%, $P = 3.1 \times 10^{-5}$, Fisher’s exact test).”

- The TF binding analysis was performed as an independent analysis assessing all SNPs for which allelic imbalance was assessed. The paragraph describing this analysis was placed before the allelic imbalance analysis (page 5 of the revised version of the manuscript).
- Method section:
 - The section describing the allelic imbalance was modified (page 9-10 of the new version of the manuscript).
 - The section “Assignments of genotypes” was modified, now including the description of the background set of SNPs used for the beta-binomial test (page 8-9 of the new version of the manuscript).
- Supplement: The results presented in supplementary table 9 (old table 6) and 11 (old table 8) were changed.

Minor points:

4. It would be nice to include specific examples/figures of ATAC-seq and DNA-methylation QTLs which show differences in signal associated with genotype.

Answer:

We thank the reviewer for this comment and we agree that including specific examples in a figure in the manuscript is of additional value. Therefore we have included a figure showing the QTL and three-dimensional chromatin interactions for the rs11637565 (TLE3) and the rs2511713 (UBR5) locus.

Changes to the manuscript:

- Figures: Figure 4 was added.

Reviewers' comments:

Reviewer #1 (Remarks to the Author):

Unfortunately, the authors have not taken up my suggestion to follow up a few interesting regulatory SNPs with fine-mapping and in-depth experimental analysis.

As it stands, the study remains descriptive and somewhat incremental, and the title claim that the study "refines regulatory mechanisms at chronic lymphocytic leukemia risk loci" is not supported by the current manuscript.

The statistical analysis remains rather hand-wavy. As mentioned in my initial review, I have no doubt - given prior research by others - that there is a cell type specific enrichment. Nevertheless, the statistical analysis would still need to be more rigorous throughout, and in particular needs to do a better job controlling for confounding factors in the genome-wide distribution of GWAS hits as well as of regions identified through epigenetic analysis.

While it will be an editorial decision whether or not this study meets the standards for depth and novelty of Nature Communications, at least the following points should be addressed in a revised version or for submission elsewhere:

1. Any claims about "mechanisms" or "functional" should be removed from the title and abstract, presenting the study more clearly as an exploratory analysis of candidate associations (not: "candidate molecular mechanisms") in CLL predisposition.
2. In-depth statistics review would be needed to iron out various issues with the use and documentation of statistical methods in the current study. For example, in the Reporting Summary, various boxes are checked as "confirmed" but are not implemented in the manuscript (e.g. description of covariates, test assumptions, confidence intervals)
3. Given the concerns about the rigor of the statistical analysis, it is particularly important that the authors make the full analysis source code needed for reproducing all figures/tables available as a supplementary file. This will help not only the reader but also the statistics reviewer.

Reviewer #3 (Remarks to the Author):

The authors have addressed all of the concerns that I raised and I appreciate the effort they made to do so.

Request 1

The statistical analysis remains rather hand-wavy. As mentioned in my initial review, I have no doubt - given prior research by others - that there is a cell type specific enrichment. Nevertheless, the statistical analysis would still need to be more rigorous throughout, and in particular needs to do a better job controlling for confounding factors in the genome-wide distribution of GWAS hits as well as of regions identified through epigenetic analysis.

Answer:

We agree with the reviewer that the background distribution of the chromatin states could be different when taking into account only the SNPs that were used in previous GWAS studies to define the 42 risk loci, while in the current approach we use the complete genome distribution as background.

We addressed this issue by taking all the linkage disequilibrium regions (LD, $r^2 \geq 0.2$) of the SNPs used in the previous GWAS studies. Next, the distribution of the chromatin states in these regions was used as the background to calculate chromatin state enrichments at CLL risk loci. This led to the exclusion of the centromeres and telomeric regions where no SNPs can be found and as expected to a decrease in heterochromatic regions in the background regions:

	CLL12	CLL182	CLL3	CLL1525	CLL1532	CLL110	CLL1228
ActProm	1.06634216	1.06603397	1.0657702	1.06626401	1.06622898	1.06621654	1.06609913
WkProm	1.06634919	1.06601428	1.0662261	1.06603543	1.06610342	1.06601629	1.06576347
PoisProm	1.06369932	1.06472028	1.0647569	1.01102961	1.0612659	1.0650244	1.06585516
StrEnh1	1.06619553	1.06648327	1.06645424	1.06613388	1.06650687	1.06646325	1.06648812
StrEnh2	1.06616203	1.06617614	1.0659417	1.06639206	1.06636131	1.06608036	1.06620956
WkEnh	1.06639621	1.06626536	1.0663073	1.06642732	1.06637436	1.06601756	1.0663206
Txn_Trans	1.0635783	1.06643472	1.06629162	1.06646653	1.06643516	1.06641504	1.06644465
Txn_Elong	1.06641917	1.06646665	1.06649234	1.0664501	1.06649898	1.06644635	1.0664442
Wk_Txn	1.06643396	1.06610736	1.06630174	1.06632027	1.06626406	1.06604831	1.06604292
H3K9me3_Repr	1.02422896	1.03722372	1.04397445	1.03747515	1.03474921	1.0489465	1.04091143
H3K27me3_Repr	1.06624484	1.0661778	1.06594385	1.06431068	1.06593432	1.06624	1.06617964
Het_LowSign	0.98684028	0.97924525	0.98373483	0.98723253	0.98429855	0.97936408	0.97795757

Table 1. Ratio of the fraction of each specific chromatin state in the new versus the old background distribution for all 7 CLL patients.

The downstream analysis led to a small change in median fold change enrichments.

Change to the manuscript:

- **Main text:** The sentence “Chromatin state enrichment in CLL essentially defines four groups of risk loci characterized by: (i), active regulatory elements and transcribed regions ($n = 13$ loci); (ii), active regulatory elements only ($n = 15$ loci); (iii), active and weak enhancers ($n = 10$ loci); and (iv), poised and inactive regions ($n = 4$ loci) (Fig. 1a)” was replaced for “Chromatin state enrichment in CLL essentially defines five groups of risk loci characterized by: (i), active regulatory elements only and transcribed regions ($n = 153$ loci); (ii), active regulatory elements only and transcribed regions ($n = 105$ loci); (iii), active and weak enhancers weak regulatory elements only ($n = 10$ loci); and (iv), weak regulatory elements and transcribed regions ($n = 3$ loci); and (v), poised and inactive regions ($n = 4$ loci) (Fig. 1a)” (end of first paragraph of the results section).

- **Method section:** In the paragraph entitled “*Linking risk loci to chromatin states*”, the following text was added: “*For each CLL sample the chromatin state enrichment at a CLL risk region, E, was calculated as*

$$E_{ij} = \log_2 \frac{C_{j;i}}{C_{j;B}}$$

where *i* is a CLL risk allele, *j* is a chromatin state, $C_{j;i}$ is the fraction of the LD region ($r^2 \geq 0.2$) of *i* covered by chromatin state *j*, and $C_{j;B}$ is the fraction of the background regions covered by chromatin state *j*. These background regions were determined as the merged LD regions ($r^2 \geq 0.2$) of all SNPs in the 1000 Genomes Project and UK10K³⁴, excluding regions containing CLL, BC or CRC risk loci (Supplementary Table 12).”

- **Figure 1:** Panels 1a and 1b were updated and the figure legend was updated accordingly. The rounded p-values associated to panel 1b did not change in the new calculations, but were corrected by Bonferroni multiple testing correction.
- **Supplementary Table 3:** The median log2FC levels were updated (column F-Q) and the standard deviations of the log2FC levels in 7 CLL cases were added (column R-AC). The supplementary table legend was updated accordingly.
- **Supplementary Table 12:** This table was added to show the genomic regions that were used in the background set.

Request 2

1. Any claims about “mechanisms” or “functional” should be removed from the title and abstract, presenting the study more clearly as an exploratory analysis of candidate associations (not: “candidate molecular mechanisms”) in CLL predisposition.

Answer:

We agree with the reviewer that the title and the abstract may be misleading, as a reader may interpret that our study includes mechanistic/functional experiments. To make this point clearer we followed the suggestions of the reviewer by changing the title and the abstract, as described below.

Change to the manuscript:

- **Title:** we changed the title from “*Integration of genomic data refines regulatory mechanisms at chronic lymphocytic leukemia risk loci*” to “*Integration of genomic data refines the link between genetic predisposition and the regulatory landscape at chronic lymphocytic leukemia risk loci*”
- **Abstract:** we changed the sentences “*At 14 loci the analysis highlighted 63 variants as the potential functional basis of CLL risk. In summary, our integrated genetic and epigenetic approach has revealed new candidate molecular mechanisms underlying CLL predisposition*” to “*At 14 loci our analysis highlights 63 variants that potentially influence CLL risk. In summary, the integration of genetic and epigenetic information has provided insights into the relationship between inherited predisposition and the regulatory landscape of CLL.*”

Request 3

2. In-depth statistics review would be needed to iron out various issues with the use and documentation of statistical methods in the current study. For example, in the Reporting Summary, various boxes are checked as “confirmed” but are not implemented in the manuscript (e.g. description of covariates, test assumptions, confidence intervals)

Answer:

We agree with the reviewer that in some cases the requested information is lacking. We went through all the analysis performed in the paper, described the missing information below and made changes to the manuscript accordingly.

Chromatin state enrichment analysis (Figure 1a and 1b, Supplementary Table S3)

1. Description of all covariates tested.
 - The percentages of CLL-, CRC- and BC-associated SNPs enriched for active chromatin in CLL were compared (fig 1b), no covariates were tested.
2. A description of any assumptions or corrections, such as tests of normality and adjustment for multiple comparisons.
 - The non-parametric wilcoxon rank sum test was used, which does not require prior test assumptions. Multiple testing was performed using the bonferroni multiple testing correction.
3. A full description of the statistics including central tendency (e.g. means) or other basic estimates (e.g. regression coefficient) AND variation (e.g. standard deviation) or associated estimates of uncertainty (e.g. confidence intervals).
 - The statistics of the datasets are included in the legend of figure 1b, including mean, median, minimum, maximum and number of datapoints. In addition, in supplementary table 3 the standard deviations of the chromatin state enrichment were added.
4. For null hypothesis testing, the test statistic (e.g. F, t, r) with confidence intervals, effect sizes, degrees of freedom and P value noted.
 - In the legend of figure 1b, the test statistic (W), effect sizes (median difference), confidence interval and corrected P-values are listed as well as the sample sizes (i.e., number of datapoints) per group. The Wilcoxon rank sum test does not require degrees of freedom.

H3K27ac enrichment analysis (Figure 1c, Supplementary Table S4)

1. Description of all covariates tested.
 - The data was corrected for the SPOT (signal proportion of tags) score. This score represents the signal-to-noise ratio of the H3K27ac ChIP-seq experiments and was previously found to be a confounding factor (Beekman et al. Nat Med 2018).
2. A description of any assumptions or corrections, such as tests of normality and adjustment for multiple comparisons.
 - DESeq2 assumes that the raw counts follow a Negative Binomial distribution and that the parameter estimations follow a normal distribution. Results were adjusted for multiple testing using the Benjamini-Hochberg procedure.
3. A full description of the statistics including central tendency (e.g. means) or other basic estimates (e.g. regression coefficient) AND variation (e.g. standard deviation) or associated estimates of uncertainty (e.g. confidence intervals).
 - The log₂FC is the regression coefficient and the variation is the SE (lfcSE). Both are estimated by the DESeq2 methodology and shown in the updated version of Supplementary Table S4.
4. For null hypothesis testing, the test statistic (e.g. F, t, r) with confidence intervals, effect sizes, degrees of freedom and P value noted.
 - The test statistic is the Wald statistic, the effect size is the log₂FC, the 95% confidence interval is calculated using the log₂FC and the lfcSE (log₂FC +/- 1.96*lfcSE), and P-values and adjusted P-values from the Wald test are listed. All this information can be found in the updated version of Supplementary Table S4. Degrees of freedom are not applicable.

QTL analysis (Figure 2, Supplementary Table S7)

1. Description of all covariates tested.
 - We used PEER to identify covariates/unknown confounders and PEER calculated residuals were used for QTL analyses.
2. A description of any assumptions or corrections, such as tests of normality and adjustment for multiple comparisons.
 - FDR is already included in results tables.
3. A full description of the statistics including central tendency (e.g. means) or other basic estimates (e.g. regression coefficient) AND variation (e.g. standard deviation) or associated estimates of uncertainty (e.g. confidence intervals).
 - Beta values already included in results tables.
4. For null hypothesis testing, the test statistic (e.g. F, t, r) with confidence intervals, effect sizes, degrees of freedom and P value noted.
 - Beta, t-statistic, P value and FDR are already included in results tables – this is the full Matrix eQTL output

Transcription factor binding analysis (Figure 3, Supplementary Table S8)

1. Description of all covariates tested.
 - NA
2. A description of any assumptions or corrections, such as tests of normality and adjustment for multiple comparisons.
 - NA
3. A full description of the statistics including central tendency (e.g. means) or other basic estimates (e.g. regression coefficient) AND variation (e.g. standard deviation) or associated estimates of uncertainty (e.g. confidence intervals).
 - Scores were calculated using the relative entropy algorithm. P-values were calculated using the score, given the score distribution using the position weight matrices.
4. For null hypothesis testing, the test statistic (e.g. F, t, r) with confidence intervals, effect sizes, degrees of freedom and P value noted.
 - NA

Allelic imbalance analysis (Supplementary Table S9)

1. Description of all covariates tested.
 - Different confounding factors (PCR amplification, reference allele mapping bias and genotyping errors) as well as exclusion of regions with potential mono-allelic expression (HLA, IGK, IGL, IGH and regions surrounding imprinted genes) were taken into account during pre-processing of the data. No covariates were tested when applying the beta-binomial test.
2. A description of any assumptions or corrections, such as tests of normality and adjustment for multiple comparisons.
 - The number of read counts containing the reference allele were assumed to follow a beta-binomial distribution with parameter n = total read counts at the corresponding position. Results were adjusted for multiple testing using the Benjamini-Hochberg procedure.
3. A full description of the statistics including central tendency (e.g. means) or other basic estimates (e.g. regression coefficient) AND variation (e.g. standard deviation) or associated estimates of uncertainty (e.g. confidence intervals).

- The alpha and beta of the beta-binomial distribution are reported. The central tendencies of the allelic imbalance analysis are represented by the allele ratio of the reference allele and the variation is represented by the 95% Wald confidence interval.
- 4. For null hypothesis testing, the test statistic (e.g. F, t, r) with confidence intervals, effect sizes, degrees of freedom and P value noted.
 - The test statistic is the counts in the reference allele, the effect size is the deviation of the central tendency from 0.5, for the confidence interval see point 3. Degrees of freedom are not relevant for this analysis. P-values and FDR values are noted.

Promoter Capture HiC analysis (Figure 4, Supplementary Table S10)

1. Description of all covariates tested.
 - NA
2. A description of any assumptions or corrections, such as tests of normality and adjustment for multiple comparisons.
 - NA
3. A full description of the statistics including central tendency (e.g. means) or other basic estimates (e.g. regression coefficient) AND variation (e.g. standard deviation) or associated estimates of uncertainty (e.g. confidence intervals).
 - We used CHICAGO to identify significant interactions. As advocated, CHi-C interactions with a score ≥ 5.0 were considered to be statistically significant [Cairns, J. et al. CHICAGO: robust detection of DNA looping interactions in Capture Hi-C data. *Genome Biol* 17, 127 (2016)].
4. For null hypothesis testing, the test statistic (e.g. F, t, r) with confidence intervals, effect sizes, degrees of freedom and P value noted.
 - NA

Change to the manuscript:

- Method section: Within the section “*Linking risk loci to chromatin states*”, the following text “*The enrichment of each chromatin state per CLL sample was calculated by dividing the proportion of the LD region for a given chromatin state, by the proportion of the total genome covered by this chromatin state.*” was replaced for: “*For each CLL sample the chromatin state enrichment at a CLL risk region, E, was calculated as*

$$E_{ij} = \log_2 \frac{C_{j,i}}{C_{j,B}}$$

where i is a CLL risk allele, j is a chromatin state, $C_{j,i}$ is the fraction of the LD region ($r^2 \geq 0.2$) of i covered by chromatin state j, and $C_{j,B}$ is the fraction of the background regions covered by chromatin state j. These background regions were determined as the merged LD regions ($r^2 \geq 0.2$) of all SNPs in the 1000 Genomes Project and UK10K³⁴, excluding regions containing CLL, BC or CRC risk loci (Supplementary Table 12). The difference in chromatin state enrichment for CLL risk loci with BC and CRC risk loci was calculated using the Wilcoxon rank sum test. No covariates were tested. To account for multiple testing a Bonferroni correction was applied.”

- Method section: Within the section “*Comparison of H3K27ac signal among CLL and normal B cells*”, the following text “*The analysis was performed by contrasting CLL with each of the normal subpopulations samples using the nbinomWaldTest*” was changed to: “*The analysis was performed by contrasting CLL with each of the normal subpopulations samples using the nbinomWaldTest, correcting for the SPOT score;. per sample, its condition (CLL or the*

- corresponding normal B-cell subpopulation) and the SPOT score were introduced into the model as per Beekman et al¹².”*
- Method section: Within the section “Allelic imbalance” the sentence “The central tendency of allelic imbalance was defined per SNP by calculating the allele ratio of the reference allele, while its 95% confidence intervals was calculated using the normal approximation.” was added.
 - Method section: Within the section “Quantitative trait locus analysis” the sentence “QTL analyses were conducted using Matrix eQTL” was modified to “Inverse normal transformed PEER computed residuals were used as input for QTL analyses which were conducted using Matrix eQTL, under a linear model.”
 - Method section: Within the section “Transcription factor motif and binding analysis” the sentence “Scores were calculated on the basis of relative entropy” was modified to “Scores were calculated using the relative entropy algorithm. P-values were calculated using the score, given the score distribution using the position weight matrices”.
 - Supplementary Table 3: The standard deviations of the chromatin state enrichment analysis was added. The legend of this table was changed accordingly.
 - Supplementary Table 4: Extra columns with the standard error and confidence interval of the log₂ FC as well as the Wald statistic were added. The legend of this table was changed accordingly.
 - Supplementary Table 9: Extra columns were added to describe the central tendency (allele ratio of reference allele) and their confidence intervals (95% Wald confidence interval). The legend of this table was changed accordingly and contains furthermore an explanation how the effect size can be obtained.

Request 4

3. Given the concerns about the rigor of the statistical analysis, it is particularly important that the authors make the full analysis source code needed for reproducing all figures/tables available as a supplementary file. This will help not only the reader but also the statistics reviewer.

Answer:

We do provide the source code for all the parts that were not directly implemented in standard packages. On top of that we have provided a website (<http://resources.idibaps.org/paper/integration-of-genomic-data-refines-the-link-between-genetic-predisposition-and-the-regulatory-landscape-at-chronic-lymphocytic-leukemia-risk-loci>) where the source codes of these parts and additional information can be found, such as the normalized data that was used as input for the QTL analysis.

- Chromatin state enrichment analysis
 - Source code provided
- H3K27ac enrichment analysis
 - Source code provided
- QTL analysis (H3K27ac, ATAC-seq, DNA methylation and RNA array data)
 - Matrix eQTL (linear model) with PEER generated residuals
 - Normalized data matrices provided
 - All published packages/methods, no source code provided
- Transcription factor binding analysis
 - MotifbreakR published package, no source code provided
- Allelic imbalance analysis

- Mapping bias filtering implemented with the standard WASP package, no source code provided
 - Source code for the downstream statistical analysis provided
 - For ethical reasons and in agreement with the policies of the International Cancer Genome Consortium (ICGC) and the BLUEPRINT Consortium, the raw ATAC-seq data and the whole genome sequencing and SNP6.0 array data to extract genotypes has to be requested via the data access committees of these consortia. For the genotype data we provide an example genotype file that can be used to test the code.
- Promoter Capture HiC analysis
- Bowtie2, HiCUP and CHiCAGO
 - All published software, no source code provided

Change to the manuscript:

Method section: Within the data availability statement, we have added the following sentence: “Furthermore, we have created a website (<http://resources.idibaps.org/paper/integration-of-genomic-data-refines-the-link-between-genetic-predisposition-and-the-regulatory-landscape-at-chronic-lymphocytic-leukemia-risk-loci>) where the source code for all the parts that were not freely available as well as additional information can be found.”

REVIEWERS' COMMENTS:

Reviewer #3 (Remarks to the Author):

The authors have already addressed all of the concerns I had with the manuscript. In my opinion, this revision also addresses the concerns raised by the other reviewer regarding descriptions of statistical methods and availability of source code.

Comment 1, reviewer 3. The authors have already addressed all of the concerns I had with the manuscript. In my opinion, this revision also addresses the concerns raised by the other reviewer regarding descriptions of statistical methods and availability of source code.

We thank reviewer 3 for the positive evaluation of the second revised version of the manuscript.